# Staying Active under Restrictions: Changes in Type of Physical Exercise during the Initial COVID-19 Lockdown

**DOI:** 10.3390/ijerph182212015

**Published:** 2021-11-16

**Authors:** Valentin Benzing, Sanaz Nosrat, Alireza Aghababa, Vassilis Barkoukis, Dmitriy Bondarev, Yu-Kai Chang, Boris Cheval, Muhammet Cihat Çiftçi, Hassan M. Elsangedy, Maria Luisa M. Guinto, Zhijian Huang, Martin Kopp, Hafrún Kristjánsdóttir, Garry Kuan, Luca Mallia, Dadi Rafnsson, Gledson Tavares Amorim Oliveira, Arto J. Pesola, Caterina Pesce, Noora J. Ronkainen, Sinika Timme, Ralf Brand

**Affiliations:** 1Institute of Sport Science, University of Bern, 3012 Bern, Switzerland; noora.ronkainen@unibe.ch; 2Department of Health Sciences, Lehman College, The City University of New York, New York, NY 10468, USA; sanaz.nosrat@lehman.cuny.edu; 3Department of Sport Psychology, Sport Sciences Research Institute (SSRI), Tehran 1587958711, Iran; alirezaaghababa@yahoo.com; 4Department of Physical Education and Sport Science, Aristotle University of Thessaloniki, 54124 Thessaloniki, Greece; bark@phed.auth.gr; 5Faculty of Sport and Health Sciences, University of Jyvaskyla, 40014 Jyvaskyla, Finland; Dmitriy.d.bondarev@jyu.fi or; 6Faculty of Sport and Health Sciences, Immanuel Kant Baltic Federal University, 236041 Kaliningrad, Russia; 7Department of Physical Education and Sport Sciences, National Taiwan Normal University, Taipei 106209, Taiwan; yukaichangnew@gmail.com; 8Institute for Research Excellence in Learning Science, National Taiwan Normal University, Taipei 106209, Taiwan; 9Swiss Center for Affective Sciences, University of Geneva, 1202 Geneva, Switzerland; Boris.Cheval@unige.ch; 10Laboratory for the Study of Emotion Elicitation and Expression (E3Lab), Department of Psychology, University of Geneva, 1211 Geneva, Switzerland; 11Department of Sport Management, Faculty of Sport Science, Ankara Yıldırım Beyazıt University, Ankara 06010, Turkey; mcciftci@ybu.edu.tr; 12Department of Physical Education, Federal University of Rio Grande do Norte, Natal 59092-050, Brazil; Hassan.elsangedy@gmail.com (H.M.E.); gledsontavares12@gmail.com (G.T.A.O.); 13Department of Sports Science, College of Human Kinetics, University of the Philippines Diliman, Quezon City 1808, Philippines; mmguinto1@up.edu.ph; 14Department of Physical Education, Hubei University, Wuhan 430069, China; zhijian.huang@gmail.com; 15Department of Sport Science, University of Innsbruck, 6020 Innsbruck, Austria; Martin.Kopp@uibk.ac.at; 16Physical Activity, Physical Education, Sport and Health Research Centre (PAPESH), Sports Science Department, School of Social Sciences, Reykjavik University, 102 Reykjavik, Iceland; hafrunkr@ru.is; 17Exercise and Sports Science Programme, School of Health Sciences, Universiti Sains Malaysia, Kubang Kerian 16150, Kelantan, Malaysia; garry@usm.my; 18Department of Movement, Human and Health Sciences, University of Rome “Foro Italico”, 00135 Rome, Italy; luca.mallia@uniroma4.it (L.M.); caterina.pesce@uniroma4.it (C.P.); 19Department of Psychology, School of Social Sciences, Reykjavik University, 101 Reykjavik, Iceland; dadira@ru.is; 20Active Life Lab, South-Eastern Finland University of Applied Sciences, 50100 Mikkeli, Finland; Arto.Pesola@xamk.fi; 21Sport and Exercise Psychology, University of Potsdam, 14469 Potsdam, Germany; stimme@uni-potsdam.de (S.T.); ralf.brand@uni-potsdam.de (R.B.)

**Keywords:** physical activity, inactivity, coronavirus, lockdown, stay-at-home, structured exercise, risk factors

## Abstract

The COVID-19 pandemic and the associated governmental restrictions suddenly changed everyday life and potentially affected exercise behavior. The aim of this study was to explore whether individuals changed their preference for certain types of physical exercise during the pandemic and to identify risk factors for inactivity. An international online survey with 13,881 adult participants from 18 countries/regions was conducted during the initial COVID-19 related lockdown (between April and May 2020). Data on types of exercise performed during and before the initial COVID-19 lockdown were collected, translated, and categorized (free-text input). Sankey charts were used to investigate these changes, and a mixed-effects logistic regression model was used to analyze risks for inactivity. Many participants managed to continue exercising but switched from playing games (e.g., football, tennis) to running, for example. In our sample, the most popular exercise types during the initial COVID-19 lockdown included endurance, muscular strength, and multimodal exercise. Regarding risk factors, higher education, living in rural areas, and physical activity before the COVID-19 lockdown reduced the risk for inactivity during the lockdown. In this relatively active multinational sample of adults, most participants were able to continue their preferred type of exercise despite restrictions, or changed to endurance type activities. Very few became physically inactive. It seems people can adapt quickly and that the constraints imposed by social distancing may even turn into an opportunity to start exercising for some. These findings may be helpful to identify individuals at risk and optimize interventions following a major context change that can disrupt the exercise routine.

## 1. Introduction

On 11 March 2020, the Director-General of the World Health Organization (WHO) declared the outbreak of the COVID-19 pandemic [1]. To contain the rapid increase in COVID-19 incidence rates, governments worldwide imposed restrictive measures that massively curtailed public and private life [2]. Aimed at reducing incidence rates [3], restrictions (e.g., closed schools, sports clubs, gyms, recreational facilities, and parks), and related insecurities (e.g., the fear of contracting the virus) created an opportunity to study behavioral adaptations in this unprecedented situation.

Many researchers assumed a decrease in physical activity (PA) and exercise [4,5,6,7], and the WHO readily advised people to stay physically active at home or outside as much as possible [8]. For this study, PA is defined as “any bodily movement produced by skeletal muscles that result in energy expenditure” [9] and exercise is a subset of PA [9] and it was broadly defined as “any activity the participants choose to do as their exercise (e.g., workouts at home, running outside, etc.)”.

PA is associated with physical and mental health [10,11] and a recent study showed that physical inactivity and poor physical fitness are associated with a higher risk for severe COVID-19 outcomes [12]. According to the WHO guidelines, adults between 18 and 65 years are recommended to participate in at least 150 min of moderate-intensity, or 75 min of vigorous-intensity aerobic PA, or a combination of both per week [13,14]. Not surprisingly, systematic reviews reported decreased PA volume and increased physical inactivity during the first COVID-19 lockdown (Note: In this article, “lockdown” refers to the bundles of governmental policies issued to reduce further spread of the virus. Examples include stay-at-home requirements, school closures, and social distancing measures) for most of the participants [7,15,16].

However, the focus of these systematic reviews was on PA volume (i.e., the frequency and duration) more than on anything else. While PA volume is generally associated with physical health [14], other PA characteristics may be as essential and as relevant to different dimensions of health [17,18]. According to the WHO definition, health is “a state of complete physical, mental, and social well-being and not merely the absence of disease or infirmity” [19]. This definition assumes multiple dimensions of health, including physiological, psychological, and social factors (and their interactions). It is more than likely that psychological and social health dimensions are not solely affected by exercise volume but also by the intensity, frequency, and type of exercise [18,19]. Furthermore, different types of sport and exercise convey diverse values and meanings to the individual; people develop identities in their chosen activity; it is not only about physical movement but belonging, continuity, and having ‘a life project’ that brings meaning to life [20]. Therefore, specific PA characteristics may be differently affected by COVID-19 and its associated restrictions.

The pandemic provided an opportunity to explore the effects of externally imposed restrictions on adherence to and changes in the type of exercise. This is a secondary analysis of the data gathered in a larger cross-sectional study that examined changes in exercise frequency and intensity conducted during the initial COVID-19 lockdown (for detailed information on the study please see [21] and https://doi.org/10.3389/fpsyg.2020.570567 [link accessed on 1 November 2021]). The original study revealed that (a) a significant number of people who did little or no exercise before the lockdown became active during the lockdown (i.e., started some type of exercise), and (b) about 1/3 of the participants lowered exercise intensities (30.2%) and shortened exercise durations (31.4%) during the initial lockdown [21]. Based on these findings, the questions arise, (a) what type of exercises were performed before and during the initial COVID-19 lockdown, and (b) what were the risk factors for inactivity during the initial COVID-19 lockdown? These questions were not addressed in the original study because to provide answers extensive coding and categorization of free-text inputs were required.

Therefore, the main aim of this study was to investigate types of exercises performed before and during the initial COVID-19 lockdown and to identify risk factors for inactivity during the initial COVID-19 lockdown. Given the uncertainty and the novelty of this situation, an explorative analysis strategy was applied to examine changes in the type of exercises performed. In addition, risk factor analyses were conducted investigating potential variables related to inactivity. We hypothesized that age, education, gender, living environment, living situation, and exercise behavior before COVID-19 would be significant predictors of physical inactivity during COVID-19.

## 2. Materials and Methods

### 2.1. Design and Procedure

For the current study, data from a previously published study in which an online questionnaire was developed by the International Research Group on COVID and exercise (IRG) were used [21]. The original study used a cross-sectional design to investigate exercise behavior before and during the initial COVID-19 lockdown. The IRG members disseminated the link to an online survey via personal networks, social media, and press releases. Data were collected between 29 March 2020, and 7 May 2020, when almost all countries worldwide enforced a specific type of lockdown restriction (for more information, see [21]; for comparing severities of restrictions in countries see Table 1). For the current study, all IRG members of countries/regions that reached a sample size larger than 100 (in the original study) were asked to participate. All accepted, and thus the following countries/regions were included: Austria, Brazil, China, Finland, Germany, Greece, Iceland, Iran, Italy, Malaysia, Philippines, Russia, Spain, Taiwan, Turkey, United Kingdom, United States of America, and Switzerland.

### 2.2. Participants

The total sample included data from 14,973 adults from 18 different countries/regions. Participants were asked whether they had any symptoms or a diagnosis of COVID-19 to exclude these individuals from the statistical analyses (*n* = 1092). This resulted in a study sample of 13,881 individuals who were on average 34.4 years old (*SD* = 13.9); men (39.6%), women (59.4%), other gender identities (1.3%); from rural (18.1%), suburban (28.5%) or urban (53.1%) living environments. Many participants indicated higher levels of education and most participants were employed with full wages (38.7%) (Table 1).

### 2.3. Measurements

#### 2.3.1. Background Variables

General information on participants (age, gender, nationality, living environment, living situation, education, and employment status) was obtained using the following questionnaire items. For living environment, participants had to indicate the country in which they live and whether they live in a rural, suburban, or urban area. The living situation was examined using the following two questionnaire items: “Including yourself, how many people currently live in your household?” (Ranging from 1 to more than 4), and “how many of your household members are under the age of 18?” (Ranging from 0 to more than 3). Regarding socioeconomic status, participants were asked to indicate their level of education (ranging from less than high school to completed graduate school with a doctoral degree) and employment status (e.g., employed with wages, student, military, etc.).

#### 2.3.2. Stringency of Governmental Restrictions during COVID-19

The governmental stringency index (GSI) serves as an indicator of governmental policies and restrictions during the time of data collection. The GSI is thus nested with the respective countries. This index was taken from https://ourworldindata.org/grapher/covid-stringency-index (accessed on 1 November 2021). It is an additive score covering nine policy areas including school, workplace, public events, gatherings, public transport, information campaigns, stay-at-home, restrictions on internal movement, travel, testing, contact tracing, face covering, and vaccination policy. The score is scaled to vary from 0 to 100, with 100 indicating the most stringent policies.

Changes in GSI over time and across countries are depicted in Figure 1. The figure shows that the study was conducted at a relatively early stage of the pandemic during the initial lockdown measures. This time represented a global crisis in which participating countries/regions had mostly reached their highest GSI (including in the Taiwan region, which had relatively low overall stringency all the time; Figure 1).

#### 2.3.3. Type of Exercise

We used two questions to ask participants whether they were doing exercise at the time of data collection and before the initial lockdown. If participants affirmed either question, a follow up question asked what type of exercise they completed most frequently. Participants were instructed that exercise in this study referred to all activities that they described as “their exercise”. This very broad definition includes walks as well as fitness training, yoga, hiking, soccer, and many more, but not PAs in the context of their occupations. Participants could provide answers as free-text input, or they could leave it blank. In addition, to determine whether participants were inactive, they were asked “How often have you exercised lately (during COVID-19)?” Response options ranged from “never” to “every day”. This data has been reported previously [21] and was therefore only used to categorize participants as inactive.

Since there is currently no categorical coding system for free-text input on types of exercise, free-text input was coded and categorized into a “broad category” and a more detailed “specific category”. The categorization was done as follows: First, free-text answers were translated into English. Second, different names for the same type of exercise (e.g., go for a run, running) were listed under the same name. Third, the responsible authors of the respective countries/regions categorized each type of exercise according to the self-developed categorization table (see Appendix A: Table A1). The resulting “broad category” includes *exercise*/*sport*, *mindfulness*, and *everyday PA*. The more detailed “specific category” includes *endurance* (e.g., running, cycling), *muscular strength* (e.g., lifting weights, push-ups), *flexibility* (e.g., stretching), *athletic fitness* (e.g., rowing, athletics), *gymnastics* (e.g., gymnastics, balance beam)*, multimodal exercise* (e.g., fitness, workout), *games* (e.g., football, tennis), *fight and martial arts* (e.g., kung fu, judo), *dance* (e.g., ballet, dance)*, skilled enjoyment* (e.g., indoor climbing, surfing), *mindfulness* (e.g., yoga, tai chi), and *everyday PA* (e.g., walking, gardening work) (see Appendix A for a detailed description of categories). Fourth, the first and last author checked all categorizations and resolved dissenting categorizations by discussing with the co-authors.

### 2.4. Statistical Analyses

Statistical tests were performed using R [22]. The Sankey chart on the specific category and the broad category during and before COVID-19 was used to visualize the change in the type of exercise. To predict physical inactivity during the initial COVID-19 lockdown, a mixed-effects logistic regression model (estimated using Maximum Likelihood and a Nelder–Mead optimizer) was used. Background variables (age; gender; living environment: rural, suburban, urban; education) and type of exercise (broad category: inactive, mindfulness, everyday PA, exercise/sport) before COVID-19 were used as predictors (fixed effects) while country/region was used as a random effect (considering national differences for example in GSI). Standardized parameters were obtained by fitting the model on a standardized version of the dataset. Confidence Intervals (*CI*s) and *p*-values were computed using the Wald approximation.

## 3. Results

### 3.1. Types and Change of Exercise during the Initial COVID-19 Lockdown

During the initial COVID-19 lockdown (compared to before), on the level of the most aggregated category (broad category), there was an increase in inactive, mindfulness, and everyday PA, and a reduction in exercise/sport (see Figure 2 for Sankey chart and Appendix B, Table A2 for exact numbers). In total, 27.49% changed from one category to another from before to during the initial COVID-19 lockdown. The largest shifts (in absolute numbers) could be observed from inactive to exercise/sport and from exercise/sport to everyday PA as well as inactive. Further, from before to during the initial COVID-19 lockdown, there were about the same number of participants who reported they had recently started to get active as well as those who recently became inactive. Interestingly, in our sample, the majority of those who became active during COVID-19 reported to start with exercise/sport as opposed to everyday PA or mindfulness.

With a closer look at the 9546 individuals who were categorized in the broad categories of exercise/sport and mindfulness before COVID-19, a total of 47.66% adjusted their behavior from before to during the initial COVID-19 lockdown (see Figure 3 for Sankey chart and Appendix B: Table A3 for exact numbers). Further, the largest total increase (in numbers) during the initial COVID-19 lockdown was in everyday PA, and there was an increase in endurance-related exercises. Not surprisingly, there was also a reduction in games, dance, and martial arts.

### 3.2. Inactivitiy and Risk for Inactivity during the Initial COVID-19 Lockdown

As shown in Figure 2, about 50% of the inactive participants (before the initial COVID-19 lockdown) stayed inactive (during the initial COVID-19 lockdown). In total, 8–16% of participants from the other three broad categories (mindfulness: 10%, everyday PA: 16%, exercise/sport: 8%) became inactive during the initial COVID-19 lockdown (see Appendix B: Table A3 for exact numbers).

When having a closer look at the specific category for the 9546 individuals who were categorized in the broad categories of exercise/sport and mindfulness before COVID-19 (see Figure 3 for Sankey Chart and Appendix B: Table A4 for exact numbers), at first glance it seems that most participants in the endurance category became inactive (a total of 306 out of 783 participants became inactive). However, when considering the proportion in each category where individuals turned inactive, it was fight and martial arts, as well as games that were most affected rather than endurance (fight and martial arts: 12%, games: 11%, skilled enjoyment: 9%, dance: 8%, endurance: 8%, muscular strength: 8%, gymnastics: 7%, multimodal exercise: 7%, athletic fitness: 4%).

The results of the risk factor analysis (mixed logistic regression) show that higher education, living in rural areas, performing mindfulness, and engaging in everyday PA as well as exercise/sport before COVID-19 were predictive of a reduced risk of inactivity during the initial COVID-19 lockdown (see Table 2).

### 3.3. Focus on Continuous Exercisers and New Exercisers

When looking at those who exercised before and during the initial COVID-19 lockdown (continuous exercisers; see Figure 4 for Sankey chart and Appendix B: Table A4 for exact numbers), there was an increase in endurance, muscular strength, everyday PA, and mindfulness. Endurance “collects” exercisers from multiple categories during the initial COVID-19 lockdown.

Similarly, when looking at those who exercised during the initial COVID-19 lockdown, but not before (new exercisers), most participants performed an endurance, muscular strength, or multimodal exercise (see Table 3).

## 4. Discussion

This study investigated changes in the type of exercise before and during the initial COVID-19 lockdown and risk factors for inactivity. The main findings show that many participants managed to continue exercising even though some had to change the type of exercise they usually engaged in before the initial COVID-19 lockdown. The results illustrate which exercise types were popular during the initial COVID-19 lockdown (e.g., endurance, muscular strength, or multimodal exercise), and which types of exercise were largely affected by the restrictions such as those engaging in fight and martial arts or games. Not surprisingly, types of exercise that were largely affected by restrictions during the initial COVID-19 lockdown (first and foremost indoor exercises with close body contact) had the largest reduction rates and had the highest proportion of individuals turning inactive. Risk factor analyses revealed that personal (e.g., higher education) and environmental (e.g., living rurally) factors, and having exercised already before the pandemic, decreased the risk of physical inactivity during the initial COVID-19 lockdown.

Results from our sample show that during a time of strict governmental stringency, many participants continued exercising, and in fact, a considerable number of those who were inactive before it started to exercise. The most frequently performed exercises, which were also most popular for those who started exercising during the initial COVID-19 lockdown, included endurance, muscular strength, and multimodal exercise. However, these results must be interpreted cautiously because of potential confounders, although being well in line with results from previous studies on the effects of COVID-19, showing (a) increase in sport participation among less active groups of Austrian (Tyroleans) adults [23]; (b) changes in exercise type in athletes [24] and (c) a majority of participants performing their exercises alone [25]. It is not surprising that endurance, muscular strength, and multimodal exercise were chosen frequently. These may be performed outside with sufficient social distance and at any time, and almost anywhere, with a few pieces of equipment and minimal expertise. It therefore seems that these types of exercises were not severely affected by restrictions and could be easily adapted to adhere to the governmental stringencies (e.g., they could be performed outdoors and/or with fewer participants).

Within the broad category of exercise/sport and mindfulness, exercises that were most affected by restrictions during the initial COVID-19 lockdown are mostly performed indoors with close body contact and were therefore difficult (or prohibited) to continue during the initial COVID-19 lockdown. For these groups to continue exercising, switching to another type of exercise and adapting to the new situation was frequently necessary. Therefore, it is likely that exercise-related learned cue-behavior associations (i.e., habits) were disrupted on several levels by the COVID-19 restrictions [26]. Cue-behavior associations and their antecedents such as consistency are essential factors in predicting change [27]. Consistency can be defined as a temporal practice structure that helps maintain a routine as it creates a protected time by providing predictability [28]. Indeed, when looking at a previous study investigating predictors of becoming inactive and reducing PA during the initial COVID-19 lockdown, identity and habit were the two most prominent predictors which distinguished between activity profiles [29]. Since disruption and change in stable contexts make behavior and habits more difficult to sustain [30]; in the future, it may be beneficial to guide those exercisers most affected by restrictions on how to maintain and form new habits to avoid inactivity. On the one hand, this could involve reverting to old habits as much as possible. A similar type of exercise (e.g., an alternative without physical contact) can be carried out at the same time with the same group of people outdoors instead of indoors. An example of this is martial arts training that is done outside performing stick fighting (Rokushakubō) instead of full body contact. Previous results from a subsample of this data support the view that enforcing continuity in exercise types is positively associated with mood [20]. On the other hand, interventions could involve goal-setting, self-monitoring, and planning to form new habits, change the type of exercise and avoid inactivity [31].

In our sample with relatively active study participants, we observed only a slight increase in inactivity. At first glance, this finding seems contradictory to the empirical evidence, which mostly reports dramatic reductions in PA volume during the initial COVID-19 lockdown [7,15,16,25,32]. However, most of the empirical evidence refers to PA volume only. This predominant focus has become very common in the medical literature. On the one hand, this is understandable because PA volume is an important factor of physical health [14]. On the other hand, PA includes other important characteristics such as intensity, frequency, and type that are often disregarded. These different PA characteristics could be important for different health dimensions and differentially influenced by COVID-19 restrictions [18,19]. Therefore, in the current study, the types of performed exercises (including if participants became inactive) were assessed instead of volume. Assessing type instead of volume has two implications: First, since exercise is only a subset of PA [33], it is possible that these two were affected differently by the COVID-19 restrictions. It is, for example, conceivable that the restrictions (e.g., home office obligation, social distancing, etc.) reduced daily PA more strongly since active transportation constitutes an integral part of daily PA [34]. Second, a mere reduction in frequency or duration was only revealed in the current study if participants changed their type of exercise or turned completely inactive. This leads to a smaller effect being shown than if the reductions were directly measured. Further studies are therefore needed to investigate the interplay between PA characteristics during COVID-19 and their relation to different health dimensions.

The risk factor analysis revealed that a higher education, living in rural areas, and being physically active before COVID-19 decreased the risk for inactivity during the initial COVID-19 lockdown. Our finding that education and living environment are important determinants and correlates of PA replicates results from earlier studies [35,36] and ecological models [37]. For example, it has been shown that education was a correlate of moderate-to-vigorous PA during COVID-19 in Canada [29] and living in urban areas increased the risk for inactivity during COVID-19 in US adults and Croatian adolescents [38,39]. In general, individuals from urban areas are more physically active [40,41], frequently engaged in sports clubs and, due to the living area and shortage of green space, are more dependent on sport facilities. Therefore, it is not surprising that both a high education level and living in rural areas were identified to reduce risk for inactivity here as well. However, it must be considered that in the current study, the predictive strength was relatively low. Speculatively, a variety of further personal, social, and environmental variables may contribute, both additively and interactively, to explain differences in exercise behavior during the initial COVID-19 lockdown.

The finding that past behavior (e.g., inactivity before the initial COVID-19 lockdown) predicts later behavior (e.g., inactivity during the initial COVID-19 lockdown) is in line with previous meta-analyses on health behaviors [42,43]. Behavior tends to be stable over time, so past behavior often influences future behavior. This influence may be both direct (and come about more or less unintentionally) and indirect (deliberate action) [42]. Regarding PA behavior, a meta-analysis showed that prior PA accounts for a large amount of the variance of later PA behavior [43]. Interestingly, in the current study, prior exercise/sport and mindfulness behavior reduced the risk for inactivity to a larger extent than everyday PA. This may be explained by the direct effect of past behavior, a process that may reflect habit [44]. Exercises included in exercise/sport categories are mostly structured, performed at a specific time and place, and follow a certain routine (e.g., preparing clothes, getting changed, etc.). In contrast, everyday PA may be less planned and more flexible in time with no specific equipment required. A structured and similar context may help cue-behavior associations (habits) to be formed and strengthened in procedural memory [45] and act as a mediator of PA change [46]. To speculate, it may be that habits and intentions associated with exercise/sport are stronger and more resistant against disruptions than those for everyday PA. These results provide further information on potential target groups for exercise interventions in similar situations.

Finally, it’s important to understand on what basis individuals choose their type of exercise and whether the similarity between exercises facilitates a change when the usual exercise becomes restricted. Person-environment fit theory suggests that individuals strive to find a fit for consistency, certainty, and predictability [47]. Empirical studies showed that if the individual’s motives and goals fit with the affordances/incentives of an exercise, it resulted in higher well-being and increased PA [48,49]. Further, it seems likely that not only explicit motivation, but also implicit attitude [50] and automatic associations [51] (which are relevant to PA) play a role in this theory. For example, automatic associations that refer to an affective valuation of an exercise [51] are comparable for similar exercises. Speculatively, changes in the type of exercise might have been influenced by the fit between the environment (e.g., restrictions) and personal attributes (e.g., needs, values, implicit associations). Therefore, certain changes in exercise type (e.g., from everyday PA to inactive) might have been more likely than others (e.g., from exercise/sport to inactive).

## 5. Limitations

Despite the international sample including more than 13,000 participants, the innovative categorization of free-text inputs, and the interesting research results, limitations are present. First, free-text inputs have the advantage that the individual written responses can be considered; however, at some instances it is difficult to categorize all inputs correctly. For example, uncertainty could have developed due to unclear responses. Therefore, it is important to consider that these categories were created exploratively and were intended to classify the respective answers as much as possible but may have been biased at times. Second, the process of questionnaire translation was not standardized, and might have introduced bias. As mentioned previously [52], to conduct this study during the time of pronounced restrictions, it was not possible to pretest and validate the translated versions of the questionnaire. Third, we used a convenience sample. Therefore, study participants may not be representative of the population in the respective countries (or even smaller regions), which could be one reason for the sample in our study being more physically active and living in urban areas. These circumstances may cause bias and limit generalizability to other populations. Fourth, this study used a cross-sectional design and self-report data. During the pandemic, there might have been different dynamics at play, which could have been better studied by multiple repeated surveys and more objective measures. Fifth, seasonal effects might have influenced the results. Given that it was comparably warm during the time of assessment (i.e., in early 2020), it could have been easier to change exercise type compared to the cold season (e.g., during the second or third wave of lockdowns in late 2020). Sixth, our definition of exercise was broad and included different categories such as mindfulness and everyday PA which is the same as the original study [21], where it was chosen because of linguistic differences in the designation of PA, exercise, and sport. For example, the distinction between PA and exercise that is common in English does not exist in German; therefore, differences between PA, exercise, and sport, as conceptualized in English, had to be paraphrased. As the result, a relatively broad but generally understandable definition was used, and it was left open to the participants to name their preferred type of exercise through a free-text input. Seventh, although this study was able to include some participants during the lockdown, it is important to note that our sample represents only a small percentage of the total population in the participating countries. Therefore, to examine the impact of COVID-19 on different countries, data from internationally representative studies are needed.

## 6. Conclusions

The current study investigated the effect of the COVID-19 pandemic and the associated governmental restrictions on human behavior. Although, according to some published reviews and meta-analyses, PA volume seems to have decreased during the first COVID-19 related lockdown in early 2020, this study showed that many individuals changed their type of exercise, continued exercising, or even took the opportunity to start exercising. These results indicate that people can adapt quickly or even use the lockdown as an opportunity to start exercising. We found that the most restricted exercise types were associated with the largest risk of becoming physically inactive during a lockdown and that personal and environmental factors such as past exercise behavior reduced the risk of inactivity. The results of this study help identify target groups for interventions and develop appropriate programs for exercise promotion.

In summary, we conclude—that the type of exercise matters! Different exercises have various affordances/incentives, which seem relevant to different dimensions of health. Therefore, further theoretically driven studies are needed on the different exercise characteristics and their effects on multiple health dimensions. Secondly, to mitigate health risks and take advantage of the opportunities, the promotion of PA and exercise should be given an important role—especially during such unfamiliar situations.

## Figures and Tables

**Figure 1 ijerph-18-12015-f001:**
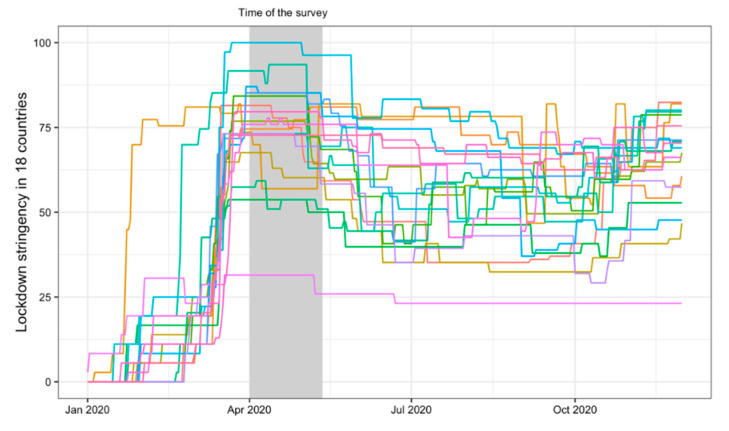
Governmental lockdown stringency index including the assessment period (Table 1 for more exact data). Note that due to the large number of countries, the data for individual countries can be viewed here: https://ourworldindata.org/grapher/covid-stringency-index (accessed on 1 November 2021).

**Figure 2 ijerph-18-12015-f002:**
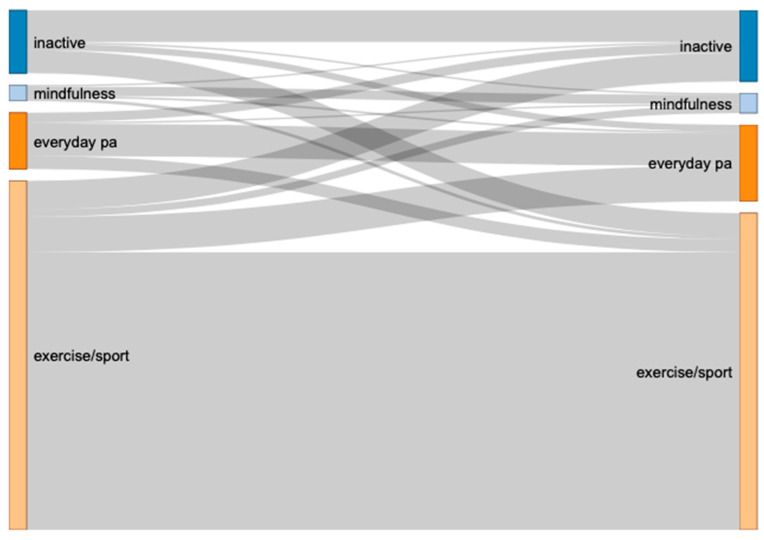
Changes in exercise types (broad category). Note that ‘before COVID-19’ is depicted on the left and ‘during the initial COVID-19 lockdown’ on the right side.

**Figure 3 ijerph-18-12015-f003:**
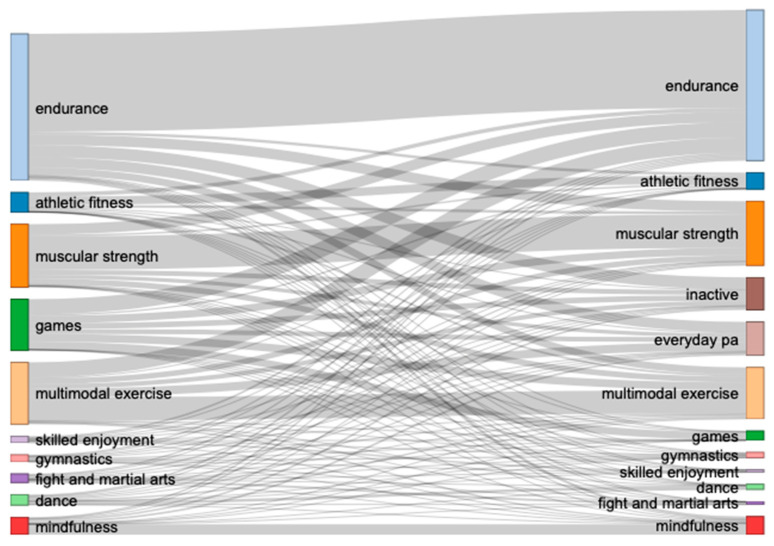
Changes in the specific category among participants that were categorized in the broad category of exercise/sport and mindfulness before COVID-19. Note that ‘before COVID-19’ is depicted on the left and ‘during the initial COVID-19 lockdown’ on the right side.

**Figure 4 ijerph-18-12015-f004:**
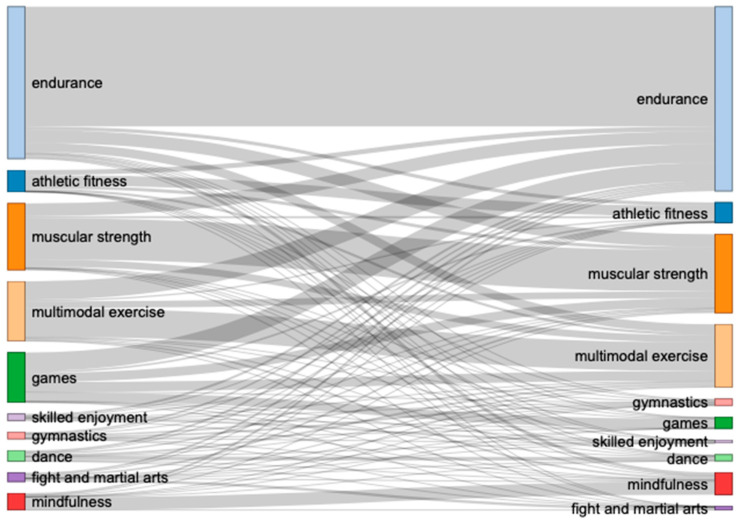
Changes in type of exercise (focus on specific category) for continuous exercisers. Note that ‘before COVID-19’ is depicted on the left and ‘during the initial COVID-19 lockdown’ on the right side.

**Table 1 ijerph-18-12015-t001:** Demographic variables and study sample characterization.

Variable	Descriptive Statistics
**Sample sizes and age**	
All participants	*n* = 13,881 (100%), *M* = 34.35 y (±13.87 y)
Male	*n* = 5449 (39.6%), *M* = 35.88 y (±14.40 y)
Female	*n* = 8248 (59.4%), *M* = 33.44 y (±13.44 y)
**Nationality and stringency of governmental containment measures ^†^**	
Austria (74.5)	*n* = 209 (56.5% female)
Brazil (76.4)	*n* = 594 (62.6% female)
China (61.1)	*n* = 815 (53.7% female)
Finland (62.9)	*n* = 471 (61.6% female)
Germany (74.5)	*n* = 2369 (61.3% female)
Greece (81.7)	*n* = 162 (57.4% female)
Iceland (52.9)	*n* = 820 (75.7% female)
Iran (54.1)	*n* = 200 (66.5% female)
Italy (86.3)	*n* = 1808 (48.0% female)
Malaysia (72.4)	*n* = 376 (61.7% female)
Philippines (98.9)	*n* = 1196 (56.2% female)
Russia (85.2)	*n* = 117 (55.6% female)
Spain (84.5)	*n* = 593 (52.6% female)
Switzerland (71.2)	*n* = 2200 (66.7% female)
Taiwan (30.8)	*n* = 1071 (53. 8% female)
Turkey (76.1)	*n* = 597 (59.3% female)
United Kingdom (79.5)	*n* = 102 (58.8% female)
United States of America (72.7)	*n* = 181 (69.1% female)
**Living environment**	
Urban	*n* = 7366 (53.1%)
Suburban	*n* = 3956 (28.5%)
Rural	*n* = 2518 (18.1%)
**Living situation**	
Living alone	1398 (10.1%)
Living with other adults (no kids)	7596 (54.7%)
Living with kids	4883 (35.2%)
**Education**	
Less than high school graduate	314 (2.3%)
High school graduate or GED	2001 (14.4%)
Some vocational school or college	1212 (8.7%)
Completed vocational school	617 (4.4%)
Completed college	2945 (21.2%)
Some graduate school	2372 (17.1%)
Graduate school: master’s degree	3413 (24.6%)
Graduate school: doctoral degree	979 (7.1%)
**Employment status**	
Employed with wages (full time)	5363 (38.7%)
Employed with wages (part time)	1467 (10.6%)
Self-employed	1103 (8.0%)
Out of work and looking for work	358 (2.6%)
Homemaker	134 (1.0%)
Student	4820 (34.8%)
Military	57 (0.4%)
Retired	432 (3.1%)
Unable to work	116 (0.8%)

Note. Missing cases or values are due to participants not providing information. ^†^ Indices calculated with data from https://ourworldindata.org/grapher/covid-stringency-index (accessed on 1 November 2021) and corresponding to national mean values during the sampling period of our study (29 March 2020 to 7 May 2020). The score is scaled to vary from 0 to 100, with 100 indicating most stringent policies.

**Table 2 ijerph-18-12015-t002:** Mixed logistic regression model predicting inactivity during the initial COVID-19 lockdown.

	Inactivity during COVID-19		
Predictors	Odds Ratios	CI	*p*
(Intercept)	0.99	0.62–1.57	0.965
Age	1.00	1.00–1.01	0.058
Education	0.92	0.89–0.95	<0.001
Gender compared to female			
male	1.01	0.90–1.14	0.808
Living environment compared to urban			
suburban	0.91	0.79–1.04	0.171
rural	0.84	0.71–1.00	0.048
Living situation compared to living with kids			
living alone	1.04	0.84–1.29	0.716
living with adult(s)	0.95	0.85–1.07	0.421
Broad category compared to inactivity before COVID-19 lockdown			
mindfulness	0.13	0.09–0.18	<0.001
everyday PA	0.27	0.23–0.18	<0.001
exercise/sport	0.12	0.11–0.14	<0.001
Random Effects			
σ2	3.29		
τ00 country/region	0.73		
ICC	0.18		
*N* country/region	18		
Observations	13,037		
Marginal R2/Conditional R2	0.121/0.281		

**Table 3 ijerph-18-12015-t003:** Types of exercises chosen by “new exercisers”.

Specific Category (*N* = 624)	Descriptive Statistics
Endurance	*n* = 243 (39%)
Muscular strength	*n* = 148 (24%)
Multimodal exercise	*n* = 130 (21%)
Games	*n* = 26 (4%)
Athletic fitness	*n* = 25 (4%)
Dance	*n* = 24 (4%)
Gymnastics	*n* = 24 (4%)
Fight and martial arts	*n* = 4 (1%)

## Data Availability

The dataset used in this study is publicly available in the online repository here: https://osf.io/qh6et/ (accessed on 1 November 2021).

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
