# Peer review of "Staying Active under Restrictions: Changes in Type of Physical Exercise during the Initial COVID-19 Lockdown"

_ijerph, 2021, doi:10.3390/ijerph182212015_

Round 1

Reviewer 1 Report

Thank you for the possibility to review this manuscript. The topic and the aim of the research is extremely important for public health. There are my comments that I believe can improve the quality of the manuscript.

Authors conclude that the risk factors for inactivity during lockdown were lower education, living in urban areas and physical inactivity before lockdown. However, Table 2 reports that higher education, living in rural areas, performing mindfulness and engaging in PA every day as well as sport/exercise before lockdown were predictive of a reduced risk of inactivity during the lockdown (lines 266-269, p. 22). However, the result that rural residency lowers the possibility of physical inactivity does not mean that urban residence increases chances for higher physical inactivity. To avoid this confusion, I suggest changing reference categories for variables. Since authors want to show risk factors for physical inactivity, I suggest to revise Table 2, to change reference categories and to show variables that increases chances of physical inactivity, but not the opposite. Or the conclusions should be revised to be in line with the results.

Since authors use term “volume of physical activity”, it should be explained in Introduction.

Reviewer 2 Report

This paper is a good work that will be interesting for the people who live there although the COVID pandemic has been interested in Italy. 

It is well written and a good methodology work. 

Thanks for giving me the opportunity to read it. 

Author Response

We thank the reviewer for the examination of our manuscript and appreciate the comments.

Reviewer 3 Report

Line 88, this is the first time 'lockdown' is mentioned. Kindly provide some context about this before bringing in this concept. 

L106, what 'original study'? A clearer explanation is needed. If this study is part of another type of study. A section about the study design is needed and a link for where further information can be found. 

The sankey figures look impressive on something like Tableau, but for scientific research, it would be important to be reinforced by the percentage. For example, in Figure 2, what percentage is in mindfulness? how much large is inactive from before to during? Each sankey figure (if remains) also needs labels from left to right. How is it in Figure 3, how is it that the data on the right is higher than on the left?

Table 2, a hypothesis for the confounders has not been stated and is needed to make sense of the variables in the analyses. 

Figure 4, why is mindfulness on the bottom on the left and second from bottom to the right. 

Table 4 appears redundant and with all the data pooled together, does not really take into consideration of the many confounders. Interpretation of data needs to be careful. 

The authors need to consider combining appendix B3 with the figures. 

Good acknowledgement of the limitations, and maybe not correct to use the term 'sizable international sample' as each country minimum require was 100. Even the smallest nation, this 100 represents less than 1% of the adult population. The criteria is too broad and biased, and should be noted as another limitation. 

L117, do not use future tense. 

L117-123 are methodological statements. 

The introduction is currently missing the main aim of the study, research questions or hypotheses. 

Table 1 is too basic and would fit better as an appendix table, or have more details about the population. One way the authors may like to consider is to stratify columns by gender or other variable. The use of the stringency measures are repeated in the figure but it is confusing as the lines move during the data collection window. 

It is not correct to say that most people completed a masters degree or higher (32%), when 38% completed college or some graduate school. This is also a very biased sample as this is not the reality of participants. 

L159-161, how were level of education and employment status used to measure SES? state explicitly if combined or used separately. 

Figure 1, each lines need labels for countries

L215-216, if the employment status was not used in the main models, either re-run them again and include it separately, or use as the SES variable, or remove from the manuscript. 

Authors need to be careful for the interpretation of the data. 

Round 2

Reviewer 1 Report

Authors addressed all suggestions and revised manuscript.